# Systems Biology Approaches Reveal Potential Phenotype-Modifier Genes in Neurofibromatosis Type 1

**DOI:** 10.3390/cancers12092416

**Published:** 2020-08-26

**Authors:** Thayne Woycinck Kowalski, Larissa Brussa Reis, Tiago Finger Andreis, Patricia Ashton-Prolla, Clévia Rosset

**Affiliations:** 1Laboratório de Medicina Genômica, Centro de Pesquisa Experimental, Hospital de Clínicas de Porto Alegre, Porto Alegre 90035-007, Rio Grande do Sul, Brazil; thaynewk@gmail.com (T.W.K.); brussareis@gmail.com (L.B.R.); tiago.f.andreis@gmail.com (T.F.A.); pprolla@hcpa.edu.br (P.A.-P.); 2Programa de Pós-Graduação em Genética e Biologia Molecular, PPGBM, Departamento de Genética, Universidade Federal do Rio Grande do Sul, Porto Alegre 91501-970, Rio Grande do Sul, Brazil; 3CESUCA - Faculdade Inedi, Cachoeirinha 94935-630, Rio Grande do Sul, Brazil; 4Serviço de Genética Médica, Hospital de Clínicas de Porto Alegre, Porto Alegre 90035-007, Rio Grande do Sul, Brazil; 5Unidade de Pesquisa Laboratorial, Centro de Pesquisa Experimental, Hospital de Clínicas de Porto Alegre, Porto Alegre 90035-007, Rio Grande do Sul, Brazil

**Keywords:** neurofibromatosis type 1, phenotype-modifier genes, systems biology

## Abstract

Neurofibromatosis type (NF1) is a syndrome characterized by varied symptoms, ranging from mild to more aggressive phenotypes. The variation is not explained only by genetic and epigenetic changes in the *NF1* gene and the concept of phenotype-modifier genes in extensively discussed in an attempt to explain this variability. Many datasets and tools are already available to explore the relationship between genetic variation and disease, including systems biology and expression data. To suggest potential NF1 modifier genes, we selected proteins related to NF1 phenotype and *NF1* gene ontologies. Protein–protein interaction (PPI) networks were assembled, and network statistics were obtained by using forward and reverse genetics strategies. We also evaluated the heterogeneous networks comprising the phenotype ontologies selected, gene expression data, and the PPI network. Finally, the hypothesized phenotype-modifier genes were verified by a random-walk mathematical model. The network statistics analyses combined with the forward and reverse genetics strategies, and the assembly of heterogeneous networks, resulted in ten potential phenotype-modifier genes: *AKT1, BRAF, EGFR, LIMK1, PAK1, PTEN, RAF1, SDC2, SMARCA4*, and *VCP*. Mathematical models using the random-walk approach suggested *SDC2* and *VCP* as the main candidate genes for phenotype-modifiers.

## 1. Introduction 

Neurofibromatosis type 1 (NF1) is a disease with a worldwide birth incidence of 1 in 2500 and a prevalence of at least 1 in 4000 [1]. The main clinical features are café-au-lait spots, axillary and inguinal freckling, cutaneous and subcutaneous neurofibromas, and Lisch nodules, occurring in almost every NF1 patient. Other less -ommon characteristics are scoliosis, macrocephaly, learning disabilities, plexiform neurofibromas, and multiple other benign and malignant tumors [2,3]. Inter-familial and intra-familial variability in NF1 is extensive: cutaneous neurofibromas may vary in number from dozens to thousands; about 30% to 50% of patients are affected by large plexiform neurofibromas; only about 10% of them develop malignant peripheral nerve sheath tumors (MPNSTs), an aggressive sarcoma and one of the most critical symptoms [2,4,5]. Other tumors outside the central nervous system occur in different frequencies between NF1 patients: low grade pilocytic astrocytomas, pheochromocytoma, gastrointestinal stromal tumor, thyroid tumors, ovary and lung tumors, breast cancer, juvenile myelomonocytic leukemia, myelodysplastic syndrome, osteosarcoma, and rhabdomyosarcoma [3,4,5].

NF1 is caused by dominant loss-of-function mutations in the tumor suppressor gene *NF1*, which encodes neurofibromin, an interactor of Ras GTPase proteins [6]. Although NF1 is a monogenic disorder of dominant character, only a few associations between a specific *NF1* variant and the disease phenotype have been reported to date. Four genotype–phenotype correlations are well described in the literature: NF1 patients harboring microdeletions have been reported to have an increased risk of malignant peripheral nerve sheath tumors, lower average intelligence, connective tissue dysplasia, skeletal malformations, and dysmorphic facial features [7,8,9]; the 3-bp in-frame deletion c.2970_2972delAA was previously associated with absence of neurofibromas [10]; the missense variant p.Arg1809Cys was associated with developmental delay and/or learning disabilities, pulmonic stenosis, and Noonan-like features, but no external plexiform neurofibromas [11]; and missense mutations affecting *NF1* codons 844–848 were associated with a more severe clinical presentation [12]. 

Apart from the aforementioned correlations, NF1patients with the same mutation may develop severe symptoms or a mild clinical expression [13,14,15]. Modifier genes, environmental factors, epigenetic factors, or a combination of them may be responsible for the remaining variability [16,17]. Modifier genes include any genes, protein-coding sequences, microRNA, and long noncoding RNA that influence one or various features of the NF1 phenotype. Primarily, modifier genes were found to be associated with phenotype variation in NF1 in large family studies, and posteriorly, NF1 animal models and knock-in and knockdown strategies have reinforced these assumptions [18]. Several strategies to discover and understand modifiers genes have been developed to help to explain the NF1 variability and were reviewed recently [19,20,21,22,23,24,25]. These strategies have identified important candidate modifier genes, and some hypotheses and associations have been established so far [16,20,22]. However, many NF1 characteristics and variability remain unexplained.

Systems biology is an integrative field that combines molecular biology experiments and computational analysis. Its aim is to understand the simplest interactions in the complexity of an organism by the evaluation of interaction networks [26,27]. By integrating genomics, proteomics, and phenotype information, it is possible to evaluate how each of these elements acts as disturber-mechanism in a specific network. This strategy consists of a very effective and economical approach to explore the disease, and might even be applied if there is little information obtained from differential gene expression studies. Hence, through systems biology tools it is possible to perform genomics research by introducing a forward or reverse strategy. The former is a strategy used by evaluating the candidate genes and how they could explain the phenotype, whilst the second strategy starts from the outcome (here NF1), and evaluates which the genes and mechanisms could be connected to it [28]. The use of a deep phenotype characterization is a good approach in conditions with heterogeneous phenotypes, when combined with next-generation sequencing data [29]. For a better comprehension of these molecular mechanisms, ontologies databases have been widely used for a correct assortment of the gene function and in the phenome characterization [30].

In this context, by using this approach, the present study searches for novel candidate NF1 modifier genes. Considering that the modifier genes could play a role in the NF1 signaling pathway or other related and unrelated pathways, in silico analyses were performed through systems biology tools involving the *NF1* gene, its protein–protein interaction network, and its related genes or phenotype ontologies. Network statistics suggested ten candidate genes and mathematical models highlighted the roles of two of them as NF1 phenotype modifiers. 

## 2. Results

A scheme presenting the main steps of the present study is available in Figure 1. To better comprehend the parameters used in each analysis, please see the Methods section.

### 2.1. Gene and Phenotype Ontologies Analyses

Gene Ontology (GO) describes a biological domain considering three aspects: molecular function, cellular component, and biological process. The Human Phenotype Ontology (HPO) provides a standardized vocabulary of phenotypic abnormalities encountered in human diseases. NF1 GO biological processes and NF1 HPO were analyzed by two coworkers individually. The chosen ontologies are listed in Appendix A.

HPO selection provided 1697 genes related to NF1 phenotype (OMIM 162200), whilst GO filter yielded 1449 genes included in the same ontologies previously selected for the *NF1* gene. When comparing both strategies, it was observed that HPO and GO analysis had 265 genes in common (Figure 2a). To assemble a network of the ontologies’ selected genes, we used the STRING tool, observing protein–protein interactions (PPI) that were previously described by experimental assays. The separate networks generated for GO and HPO analyses are represented in Appendix A, respectively. A combined network, comprising NF1 direct interactions (first neighbors) for both GO and HPO strategies is available in Figure 2b.

### 2.2. Network Statistics

To verify the nodes with relevant roles in the information flow from the network assembled in the previous section (Figure 2), systems biology network statistics were applied using Cytoscape v.3.7.2 software. Two main parameters were observed: (I) betweenness centrality, a measure based on the communication paths, meaning the nodes with high betweenness centrality could be important in the control of the information flow; and (II) the closeness centrality measure, which is based on the fastness of this information flow (from the central node to the others) [31]. The resulting network had 1561 nodes, making it difficult to visualize the main nodes. A simplified version, representing only the first neighbors, can be assessed in Figure 3.

According to this strategy, AKT1 presents the highest levels of betweenness and closeness centrality. However, despite NF1 being a highly connected protein in the network evaluated, it presented low levels of betweenness and closeness centrality, as can be observed by the node size (small) and color (light yellow, compared to the dark orange elements). Hence, we aimed to evaluate HPO and GO networks separately using the same approach to minimize the possibility of overlooking potential phenotype-modifier genes, as described in the following section.

### 2.3. Forward and Reverse Genetics Strategies

As mentioned before, GO and HPO databases are related, respectively, to the gene function and phenotype association. Hence, the observations of their independent networks, previously represented in Appendix A, were based on the forward and reverse genetics concepts. 

When evaluating betweenness and closeness centrality by the forward genetics strategy (GO network), six genes were selected: *AKT1, RAF1, LIMK1, BRAF, EGFR*, and *PTEN*. In the other analysis, the reverse genetics approach (evaluating the HPO network) provided six genes as well: *PAK1, VCP, AKT1, SMARCA4, RAF1*, and *PTEN*. Together, the strategies provided nine candidate genes for neurofibromatosis phenotype modifiers. Besides, the network communities were evaluated, and *AKT1* was identified as the network main hub, whilst the *RAF1* gene had the highest score for authority. The authority score estimates the importance of the node itself, and the hub score measures this importance based on the other nodes which are linked to the main hub. Despite the network statistics having provided these candidates, we wanted to observe whether or not their expression was altered in the absence of *NF1*. Therefore, the next step was designed to conduct gene expression analyses to evaluate this hypothesis.

### 2.4. Differential Gene Expression Networks

To comprehend the differential gene expressions (DGEs) of the candidate phenotype-modifier genes, we performed secondary expression analysis on the data available in two public repositories: The Cancer Genome Atlas (TCGA) and the Gene Expression Omnibus (GEO).

Using the GEO database, *NF1* knockdown and knockout assays were selected: GSE14038 and GSE115406. The log fold-change (logFC) and false discovery rate (FDR) values for the ten potential modifier genes in each dataset are presented in Appendix A.

In TCGA, we selected seven different types of tumors for which samples with nonsense mutations in *NF1* were available. We evaluated tumors that presented *NF1* nonsense mutations against tumors with wildtype *NF1*. The logFC and adjusted *p*-values, after the FDR correction, for the seven tumors evaluated and ten candidate genes are available in Appendix A. Despite the somatic origins of these tumors, we believe this information is valuable in order to check how NF1-loss could affect the global gene expression in a tissue/site-specific way, and check for signatures more related to a certain phenotype. 

For both TCGA and GEO assays, few genes were evidenced to be significantly differentially expressed, demonstrating that the expression of all the candidates, and for the *NF1* gene, is strictly regulated. We did not identify a variable expression profile between the tumors evaluated, and knockout assays. Therefore, we performed other systems biology analysis with the PhenomeScape application v.1.0.4 from Cytoscape software, assembling a complex network (Figure 1, step 3). For that, we used as input (I) the ontologies selected from HPO database (Figure 1, step 1); (II) the network generated in STRING tool, as also mentioned in step 1 (Figure 1); and (III) expression data from studies selected from GEO database (Figure 1, step 2). The resulting upregulated genes (overexpressed) were presented in red and the downregulated (lower expression) in green. These networks are available in Appendix A. *NF1* is downregulated (lower expression) in the knockdown and knockout studies, and upregulated (overexpressed) in the evaluation of malignant tumors when compared to benign neurofibromas. *NF1* is absent from the network when its expression is not significantly altered in the expression dataset.

Besides the genes previously selected in the forward and reverse genetics analysis, when evaluating the complex networks, we observed that the *SDC2* gene also had its expression altered when *NF1* was affected. Furthermore, *SDC2* is the first neighbor of *NF1*, which means both genes share a direct protein–protein interaction. 

### 2.5. Systems Biology Approaches Reveal 10 NF1 Phenotype-Modifier Candidate Genes 

Table 1 shows the final list of the 10 genes selected as potential phenotype-modifiers in this study and summarizes the strategies by which they were found. We then generated a complex network comprising all the candidate phenotype-modifier genes selected so far, and the NF1 phenotypes they are related to (Figure 4); the phenotypes were provided by the PhenomeScape tool, according to the data available in HPO database. Finally, we used a mathematical model to evaluate whether one of those genes could be a stronger candidate as a phenotype-modifier than the others, which is described in the following section.

### 2.6. Random Walk Analysis

A random walk is a mathematical model known as a random process. It is based on the idea that a gene (node) is an imaginary particle that performs a succession of random steps (interactions) in a network [32]. Our aim was to evaluate whether these random steps could lead the gene to the phenotype, which was set as neurofibromatosis type 1 (OMIM 162200). For this goal, we performed the random walk analysis with the *RandomWalkRestartMH* package in R v.3.6.2.

According to this mathematical model, the *NF1* gene only had to take “one step” (one interaction) to reach the phenotype OMIM 162200. The genes *SDC2* and *VCP* had to take only two steps (two interactions) (Figure 5), whilst the other eight candidates needed more interactions to cause the syndrome (Appendix A). Genes *LIMK1* and *PAK1* also needed a higher number of potential interactions, and hence more steps, than the others.

With this analysis, we confirmed all the candidates as potential phenotype-modifier genes. However, the results using this model pointed to *SDC2* and *VCP* as being more directly connected to the NF1 phenotype.

### 2.7. Literature Review and Genomic Databases Evaluation

To check for genetic variants already described in our 10 candidate genes, we looked at two databases: The Genome Aggregation Database (gnomAD v. 2.1.1), which spans 125,748 exomes and 15,708 genomes from unrelated individuals; and ClinVar, which aggregates information about genomic variation and its relationship to human health. For gnomAD, we found a total of 11,211 variants (Table 2). *SMARCA4* and *EGFR* have the highest numbers of variants (2575 and 2000, respectively); *SDC2* and *PTEN* the lower (335 and 456, respectively).

In ClinVar, germline variants were reported for all genes (N = 5216), with the exception of *SDC2*. Almost half (46.4%) of them were submitted as variants of uncertain significance (VUS). *EGFR* has the highest rate of VUS (67.3%), while all the 36 *LIMK1* variants are classified as benign/likely benign (B/LB). On the opposite way, *PTEN* and *BRAF* have the highest number of pathogenic/likely pathogenic (P/LP) variants, corresponding to 32.5% and 23.9% of all reported variants, respectively (Table 3). 

Finally, we also explored TCGA and Genomics Evidence Neoplasia Information Exchange (GENIE) datasets to check for tumor samples harboring both genetic alterations in one of our candidate genes and *NF1*. Due to lack of samples or to the higher mutational and clinical heterogeneity, we managed to make reasonable assumptions only for *AKT1, VCP*, and *SDC2*. More details are presented in the discussion section. 

## 3. Discussion

It is evident that genetic variants in *NF1* do not act alone to determinate disease phenotype. Many factors may contribute to disease variability, including environmental factors, the occurrence of epigenetic alterations, and somatic second hits in NF1-associated tumors. The accumulation of somatic NF1 mutations is much more difficult to evaluate since each tumor needs to be sequenced individually, but it may be responsible for some level of NF1 variability. Other symptoms, like delayed mental development, are less influenced by second hit mutations. Genetic modifiers in a single locus or the interaction between several genes may suppress or enhance disease severity, including genes involved in the pathways other than the NF1-Ras-mTOR pathway. There is evidence that genetic modifiers explain a major fraction of phenotypic variation in NF1 [16]. A few genes and their variants have already been described as phenotype modifiers in literature and were reviewed and summarized in Table 4, but they are still insufficient to explain all the variability found in NF1 patients. 

Recently, a review pointed out the main methods with which to discover novel phenotype-modifier genes in Mendelian diseases and formulate hypotheses about other pathways than Ras-NF1 that could be phenotype modifiers [44]. The most used methods to select candidate modifier genes are whole-genome sequencing, genome-wide association studies, and experimental approaches using animal models. Other studies also select candidate modifier genes using differential gene expression analysis [20]. These strategies have proved their value in identifying a few phenotype-modifier genes to date; however, they have some disadvantages, such as the high costs involved, being time-consuming, the use and maintenance of animal models, and the confounding factors in studies with selected NF1 patients [18]. 

One of these limitations was observed in our expression analysis, for which a few candidate genes were actually differentially expressed. Furthermore, differential expression analysis using TCGA tumor samples (Appendix A) generated distinct results when compared to GEO controlled knockdown/knockout experiments (Appendix A). Expression may depend on the tumor heterogeneity, i.e., the number of cells that are actually not expressing functional NF1, and its location, since the expression profiles of *NF1* and related genes are tissue-dependent. Gene expression is by far the most common analysis among multi-omics studies. Despite that, in many studies it is not possible to obtain a clear scenario of the biological mechanisms disrupted in a disease by evaluating only the mRNA levels. Known disease genes are often not differentially expressed in affected individuals, once the mutations may only alter the protein function or post-translational mechanisms. As a consequence, much information contained in transcriptomics datasets are ignored, demanding an alternate strategy to evaluate these multi-omics assays [45]. On the other hand, the differential gene expression networks also allowed the selection of the *SDC2* gene as a new candidate, once its expression was altered when *NF1* was affected. This example highlighted the need to evaluate the multi-omics data in a more integrated and multidisciplinary perspective [46].

Network analysis makes it possible to combine different multi-omics studies, a strategy that has been applied in several personalized medicine studies evaluating genetic syndromes with phenotypic variability [47,48,49]. Recently, many projects and consortiums were created, gathering a huge amount of public results with germline and somatic mutation databases, transcriptomics, proteomics, and metabolomics data that can now be evaluated with a systems biology tools [44]. The analysis proposed here can be seen as an optimization in the search for candidate genes acting as phenotype modifiers in NF1, which can later be confirmed by the more robust molecular and functional assays. A brief description of the potential phenotype modifiers found in this study and their variants is provided below to show mechanistic insights and facilitate experimental studies. They are also summarized in Table 1 and Table 2 and Table 3, respectively. 

The ontologies selection was an important step in our analysis, especially because *NF1* is important in several molecular mechanisms. If not filtered, our analysis could be later compromised by evidencing genes that are not so deeply associated to neurofibromatosis type 1, but which are more frequently studied in cancer (i.e., *TP53* and developmental genes). This strategy has also been applied in studies that do not identify (or do not have access to) differentially expressed genes [50,51]. For Human Phenotype Ontology (HPO), our workflow was based on a deep phenotyping strategy (computational analysis of detailed, individual clinical abnormalities) [29], according to the heterogeneity of neurofibromatosis type 1 visualized in our patients. We also aimed to avoid ontologies related to congenital anomalies outside the NF1 spectrum of phenotypes, especially the ones that could have led to embryo lethality or severe impairments (major malformations) that would have been diagnosed before NF1. Embryo development is a critical, stepwise controlled process that can be disrupted by genetic or environmental factors, such as maternal infections or exposures [52]. Since the data on *NF1* expression in the embryonic period are scarce, and we do not have information about the maternal genome or environment, we focused on the functional anomalies (more related to the fetal period) that are better characterized in neurofibromatosis type 1.

In our forward strategy, we found the epidermal growth factor receptor (EGFR) that acts upstream of NF1 in the Ras signaling pathway. EGFR belongs to a family of receptor tyrosine kinases that are anchored to the cytoplasmic membrane. EGFR is frequently over-activated in cancer and studies have shown that it is not expressed by normal Schwann cells but it is overexpressed in subpopulations of NF1 mutant Schwann cells [53]. The great involvement of this gene in different cancers and its differential expression patterns in NF1-enriched tissues may indicate that the occurrence of minor variants in this gene could act as phenotype modifiers in NF1. There are 2000 *EGFR* variants registered in the gnomAD database, 34.1% of them missense mutations. In ClinVar, most of the 199 catalogued germline variants are VUS (N = 134). Among the four P/LP variants, one (C326F) was related to Cowden syndrome. Variants in promoter and UTR regions might also have a potential as phenotype modifiers, since animal models have already shown that high levels of *EGFR* expression modify the initiation of neurofibromas, increasing their numbers [54].

*AKT1, BRAF, LIMK1, PTEN*, and *RAF1* genes were also suggested by the forward strategy. They encode proteins that act downstream of NF1 in the Ras signaling pathway. Phosphoinositide 3-kinase PI3K/AKT is one of the most frequently activated pathways in cancer. This activation may occur through mutation of multiple genes, including *PTEN, PIK3R1*, and *mTORC1* [55]. *AKT1* presented the highest levels of betweenness and closeness centrality in systems biology network analysis, demonstrating itself to be the most relevant gene in the information flux among our selected genes (Figure 3). 

*AKT1* germline mutations are mainly associated with Cowden syndrome, characterized by the appearance of hamartomas, and an increased risk of developing multiple cancers, especially breast cancer [56]. One particular pathogenic variant, E17K, is reported to be linked to 22 different conditions in ClinVar. This variant was also found in gnomAD in a European individual. To explore how this alteration could modify the phenotype when co-occurring with *NF1* mutations, we looked at mutational and clinical data deposited in the GENIE database (v7.0). When excluding *AKT1*-mutated patients and considering only *NF1* mutations with more deleterious effects (nonsense, frameshift, and splicing variants), the most frequent cancer types are non-small cell lung cancer (12%), glioma (10%), and melanoma (9%). However, when grouping samples with both *AKT1*-E17K and *NF1* mutations, breast cancer becomes the most prevalent, corresponding to 73% of all tumors. The link between breast cancer susceptibility and *NF1* alterations was already established [18]. However, the mechanism that leads to this specific phenotype remains to be elucidated and *AKT1* emerges as a strong candidate.

PI3K-Akt pathway activity is negatively regulated by phosphatase and tensin homolog protein (PTEN) [57]. PTEN is a tumor suppressor and its inactivation has a role in plexiform neurofibroma tumorigenesis and progression to high-grade peripheral nerve sheath tumors in the context of NF1 loss in Schwann cells, which is a very variable symptom in NF1, and may also participate in the mechanism of tumorigenesis of other tumors related to NF1 [58,59]. There are not many variants catalogued for *PTEN* in gnomAD (N = 456), but 1546 were already reported in ClinVar, with 35% still being classified as VUS. The phenotypes related to *PTEN* variants are, as expected, similar to *AKT1*, including Cowden syndrome and hereditary breast and ovarian cancer syndrome. Considering the importance of both *AKT1* and *PTEN* in tumorigenesis, variants in the corresponding genes, not necessarily pathogenic, could act as a modifier of NF1 disease and need to be further investigated. 

In contrast, BRAF studies in NF1 patients were already conducted. *BRAF* gene encodes a protein belonging to the RAF family of serine/threonine protein kinases. This protein plays a role in regulating the MAPK/ERK signaling pathway, which affects cell division, differentiation, and secretion. Germline mutations in *BRAF* were previously associated with cardiofaciocutaneous, Noonan, and Costello syndromes [60]. In ClinVar, 334 germline variants were already submitted, 80 being classified as P/LP and 117 of the remainder as VUS. A recent study analyzed a cohort of 100 patients clinically suspected of NF1 and identified 73 *NF1* mutations and two *BRAF* novel variants. The clinical features of NF1 patients with co-occurrence of *NF1-BRAF* mutations were severe, and *BRAF* variants may have a synergistic role in determining NF1 phenotype [61]. 

Another member of the same family, the *RAF1* gene, was suggested by the forward strategy. The *RAF1* gene encodes a serine/threonine kinase protein that functions downstream of RAS and activates MEK1 and MEK2. In GENIE, *RAF1* mutations are observed in various cancers. LEOPARD and Noonan syndromes were already associated in ClinVar with 6 and 28 pathogenic *RAF1* variants, respectively. The other 197 variants were still classified as VUS and included conditions such as other rasopathies, chordoma, and retinoblastoma. 

Finally, by the forward strategy, *LIMK1* was suggested as a phenotype modifier. The N-terminal domain of LIM kinase 1 (*LIMK1*) regulates actin dynamics, affects cell adhesion and migration by phosphorylating cofilin, and negatively regulates the Rac1/Pak1/LIMK1/cofilin pathway [62]. *NF1* is an upstream regulator of *LIMK1* by acting on cofilin phosphorylation. When *NF1* is mutated, this pathway is affected, possibly influencing neuronal development and cognitive deficits associated with the disease [62,63]. We found 1133 *LIMK1* variants in gnomAD, but only 36 reported as germline in ClinVar, all of them classified as benign/likely benign. That may indicate that mutated *LIMK1* alone is not pathogenic, but it does not exclude a combined effect of *NF1* variants acting as a phenotype modifier. 

By the reverse strategy, *AKT1, PTEN,* and *RAF1* were also suggested, reinforcing the possible roles of these genes in NF1 phenotypes. Additionally, *PAK1, SMARCA4,* and *VCP* were found. The kinase PAK1 is a Rac/CDC42-dependent serine/threonine kinase that acts by activating several kinases such as RAF, ERK, and LIMK1, and other related pathways by activating TGFα and VEGF. PAK1 is required for the malignant growth of RAS transformants in NF1 neurofibrosarcoma cell lines [64,65]. There are not many PAK1 variants registered in gnomAD (N = 774) and only six in ClinVar. However, two LP variants (Y131C and Y429C) reported in ClinVar were associated with an intellectual developmental disorder with macrocephaly, seizures, and speech delay, phenotypes that are reported in NF1 patients. 

*SMARCA4* is a central component of the switch/sucrose-non-fermentable (SWI/SNF) chromatin remodeling complex. Inactivating mutations and loss of expression in several components of this complex have been implicated in carcinogenesis [66,67,68]. Thus, variants in one of these genes might influence the NF1 phenotype. *SMARCA4* has the highest number of variants in both gnomAD and ClinVar among our candidate genes: 2575 and 2310, respectively. Peripheral nerve sheath tumors were already reported in patients with the syndrome carrying *SMARCB1* mutations, which belongs to the same family of *SMARCA4* [14]. Loss of *SMARCB1* was also related to Schwannoma, another phenotype found in NF1 despite being more frequent in NF2 [69].

The valosin-containing protein (*VCP*) appeared as a strong candidate for being an NF1 phenotype modifier by our random-walk analysis. *VCP* gene is associated with the multisystem degenerative autosomal dominant disorder of inclusion body myopathy with Paget disease of bone and frontotemporal dementia (IBMPFD) and mutations were related to 1–2% of amyotrophic lateral sclerosis cases [70]. However, missense mutations in *VCP*, and low-effect and low-penetrant mutations in this gene, have controversial roles in causing disease. Neurofibromin interacts with VCP through its Leucine- Rich Domain (LRD)-domain [71]. Patients with NF1 who have mutations in the LRD coding region were described to be more prone to developing cognitive deficits than those with mutations elsewhere in the *NF1* gene [72]. In the same study, it was observed that point mutations in the LRD coding region in the *NF1* gene abolished the ability of NF1 to interact with VCP, while *VCP* mutants were shown to have reduced affinity for NF1. Interestingly, non-disease-associated polymorphisms in the LRD region of the *NF1* gene may increase the risk of an IBMPFD patient developing dementia. In the same way, polymorphisms in the *VCP* gene that code for domains that interact with NF1 might influence the NF1 phenotype. These data obtained from literature research reinforce the accuracy of our systems biology analysis and random-walk mathematical model, which pointed *VCP* as a strong NF1 phenotype modifier candidate. 

It would not be a surprise if *VCP* variants were found co-occurring with *NF1* alterations, especially the ones in ATPAse domains 1 and 2 (D1 and D2), responsible for interacting with neurofibromin’s LRD-domain. For example, in ClinVar only two variants are reported as pathogenic D1/D2, one in each domain. On the other hand, 34 remain as VUS, 23 in D1, and 11 in D2. In gnomAD, more than half of the (N = 491) cataloged *VCP* variants are located in D1 and D2 domains. Looking at TCGA, there are few samples (N = 28) with *NF1* mutations co-occurring with variants in *VCP* D1/D2 domains, most of them (47.2%) from uterine corpus endometrial carcinomas.

The last gene that was suggested by our analysis is *SDC2*, which was found by differential gene expression networks and pointed to as a strong candidate by our mathematical model. The heparan sulfate part of SDC2 interacts with extracellular matrix proteins and growth factors to act as an adhesion molecule and as a coreceptor [73]. Variants in this gene might be associated with autism spectrum disorder [74]. Interestingly, some studies showed a higher frequency of autism spectrum disorder in NF1 children, and this is a variable condition in NF1 that might be influenced by variants in other genes [75]. Despite *SDC2* emerging as a strong candidate in our study, only 335 variants were found in the gnomAD database and none in ClinVar, suggesting a highly conserved gene. However, the lack of *SDC2* in ClinVar may merely reflect its absence from gene panels used for diagnostic purposes. On the other hand, TCGA somatic samples carrying both *NF1* and *SDC2* mutations are scarce (N = 14/10,437), most of them (57.1%) also related to uterine corpus endometrial carcinomas. This finding does not exclude the gene as a phenotype modifier, but variants co-occurring with *NF1* alterations might be a rare event. 

The literature search for variants and functions of the candidate modifier genes identified by our strategy shows that this is an economical and accurate way to filter and select genes that would be further validated by experimental assays. As a perspective, variants in the ten genes selected by our strategy will be searched in NF1 patients with different symptoms. Many strategies could be used to subsequently evaluate and validate the selected genes. One of them is to identify variants in these genes or other nearby genes and genotype those variants in NF1 patients with different symptoms and control populations, followed by statistical methods to identify correlations with the phenotype. Moreover, in-vitro and in-vivo studies are also useful for validating previously selected genes, focusing on CRISPR/Cas9 assays to induce partially and complete loss of the proteins. Our candidate genes could be also included in commercial gene panels with a low impact on their coast, which would help to feed public databases such as ClinVar. 

One obvious limitation of the present study is the lack of proper validation for the candidate phenotype-modifier genes using benchwork. Hence, the results obtained must be evaluated with caution. Experimental validation is necessary and strongly recommended before clinical extrapolation. However, our purpose was to provide a new look in the strategies for evaluation of neurofibromatosis using the huge amount of data already available in shared public-curated datasets. For example, the protein–protein interactions identified by us were previously validated by several in vitro assays and here combined in a network. Together with our network analysis, we also performed random walk, a robust mathematical model that has been applied in the analysis of biological multiplex heterogeneous networks [76,77]. We hope this complex and robust systems biology approach will help to better understand the neurofibromatosis type 1 and its phenotypic variation. 

## 4. Materials and Methods

### 4.1. Selection of NF1 Ontologies

The complete list of NF1 gene ontologies (GO) and phenotype ontologies were obtained in the AmiGO and Human Phenotype Ontology (HPO) databases, respectively. In modifier studies, the selection of which phenotypes to study is a key step, and in NF1, several phenotypic features are time-dependent [19]. Then, we selected from both lists (GO and HPO) the ontologies related to the less frequent and variable characteristics and not necessarily time-dependent, presented by NF1 patients, as reported in the literature and in our clinical experience in the Oncogenetics clinics of Hospital de Clínicas de Porto Alegre [16,78,79]. For example, cutaneous neurofibromas are common and may occur in up to 99% of NF1 patients in a cohort; this is a variable characteristic, mainly in the number of neurofibromas, but it is less variable when considering their presence in NF1 patients. Thus, we focused on ontologies related to characteristics that occur in a smaller number of patients to try to explain the variability of less common but more aggressive NF1 symptoms, such as breast cancer, delayed mental development, plexiform neurofibromas, and facial dysmorphism. The processes and phenotypes selected for the analysis involved NF1 and NF1-related signaling pathways, such as the MAPK cascade and regulation of the Ras pathway, considering the upper ontology in the hierarchy of each database. Hence, some ontologies were not selected because there was an upper term in the hierarchy that encompassed these ontologies. It is worth mentioning that we followed a guide for the correct selection of ontologies to try to limit the bias introduced by the choice of terms and keywords [80]. The processes were evaluated by two independent researchers and selected for subsequent analysis when both researchers considered it relevant. 

### 4.2. Systems Biology Analysis

Networks were generated using STRING database v.11, comprising protein–protein interactions (PPI) for *Homo sapiens*. Only experimental interactions were selected, with a minimum required interaction score set in 0.400 (default). The assembled networks were transferred to Cytoscape v.3.7.2 software, with which the network statistics was obtained. Big nodes represented proteins with high betweenness centrality scores and warm colors comprised proteins with high closeness centrality measures.

Comparison between networks was performed with DyNet application for Cytoscape v.3.7.2. Complex networks comprising HPO selected phenotypes, gene expression, and PPI networks were assembled using the PhenomeScape app, also in Cytoscape v.3.7.2, using the default settings.

### 4.3. Gene Expression Evaluation

RNA-seq and microarray secondary analysis were performed using studies selected in Gene Expression Omnibus (GEO) and The Cancer Genome Atlas (TCGA) databases. For the GEO studies, we looked for *NF1* knockdown or knockout assays and selected only the ones performed in human tumor cells. The data extraction was performed manually, and the robust multiarray averaging (RMA) normalization was applied using *oligo* or *affy* R packages (R v.3.6.2). The differentially expressed genes were obtained using the *limma* package (R v.3.6.2). 

Firstly, for TCGA, we selected seven somatic tumors with nonsense mutations in *NF1* (*NF1*-ns): bladder urothelial carcinoma (BLCA), brain lower grade glioma (LGG), breast invasive carcinoma (BRCA), cervical squamous cell carcinoma and endocervical adenocarcinoma (CESC), colon adenocarcinoma (COAD), glioblastoma multiforme (GBM), and pheochromocytoma and paraganglioma (PCPG). Despite tumors having a somatic origin, this information can be useful to check how alterations in *NF1* could affect the global gene expression in a specific tissue/tumor (phenotype). We then compared these samples against wild type *NF1* tumors to check which genes were differentially expressed only in the *NF1*-ns group. The gene expression analysis for TCGA data was performed by extracting the data with *TCGAbiolinks* package and evaluating differential gene expression with *edgeR* package. All analyses were performed in R v.3.6.2.

### 4.4. Random Walk Analysis

The heterogeneous networks comprising both the genes and phenotypes selected were assembled in the *RandomWalkRestartMH* package, in R v.3.6.2, and the random walk analysis was performed with the same package.

### 4.5. Database Research 

Other databases consulted to obtain data for the potential NF1 phenotype-modifier genes were: (1) BioGrid, for curated protein interactions; (2) PINOT tool, for literature data on curated protein interactions; (3) STRING database, for protein–protein interactions; and (4) The Human Reference Protein Interactome (HuRI), for the binary protein–protein interactions.

### 4.6. Variant Datasets

To explore variants in our candidate genes already reported in the general population or with clinical significance, we consulted The Genome Aggregation Database (gnomAD) v 2.1.1 and the ClinVar archive. For gnomAD, variants were classified according to its annotation. In ClinVar, only variants reported by at least one submitter as a germline were considered and classified according to their interpretations.

Finally, an additional analysis was performed consulting the 79,720 tumor samples made available by the AACR Project GENIE and the 10,967 samples from the TCGA PanCancer Atlas studies, using the cBioPortal for Cancer Genomics. Samples were filtered according to their mutational status: *NF1*-mutated patients including only nonsense, frameshift, and splicing; and patients with selected variants in our candidate genes, if available. Then, the clinical data were accessed and confronted with the mutational status to check which cancer types were predominant when *NF1* was exclusively altered and when *NF1* variant co-occurred with variants in our candidate genes.

## 5. Conclusions

We presented here a not yet explored systems biology strategy to investigate *NF1* phenotype modifiers. The public availability of multi-omics datasets makes possible the use of robust tools to generate complex networks including protein–protein interactions, differential expression data, and phenotypes, reinforced by mathematical models such as random-walk. Combining all these strategies, we found 10 candidate genes as potential NF1 phenotype modifiers. Resources and time may be scarce to carry out association studies and systems biology analyses makes possible to better explain the genetic heterogeneity of this complex syndrome. Our results must be interpreted cautiously in clinical application and may guide further in-vitro and in-vivo validation studies, saving time and financial resources. The approach presented here may guide further in-vitro and in-vivo validation studies, saving time and financial resources.

## Figures and Tables

**Figure 1 cancers-12-02416-f001:**
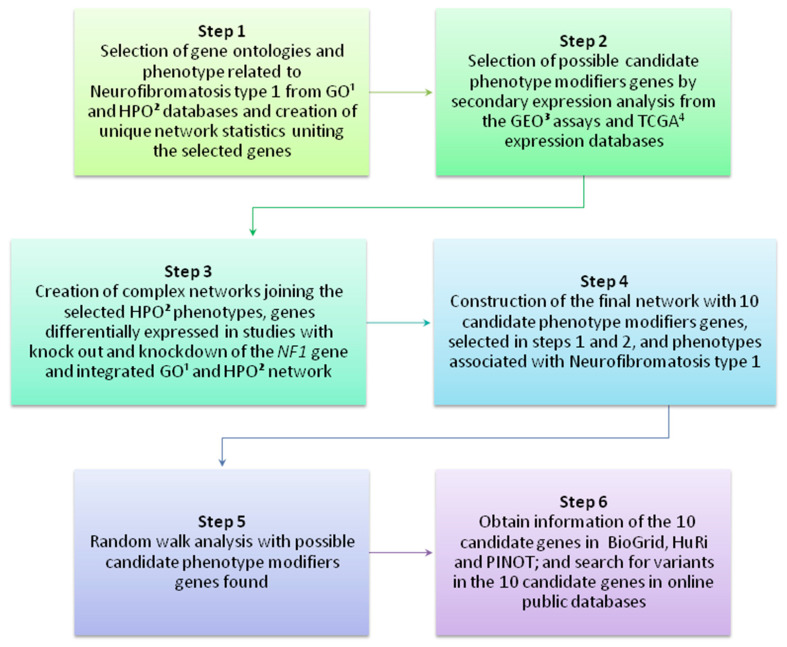
Main steps of the present study. The scheme shows the main steps in chronological order to identity potential NF1 phenotype-modifier genes. Gene Ontology (GO)^1^; Human Phenotype Ontology (HPO)^2^; Gene Expression Omnibus (GEO)^3^; The Cancer Genomic Atlas (TCGA)^4^.

**Figure 2 cancers-12-02416-f002:**
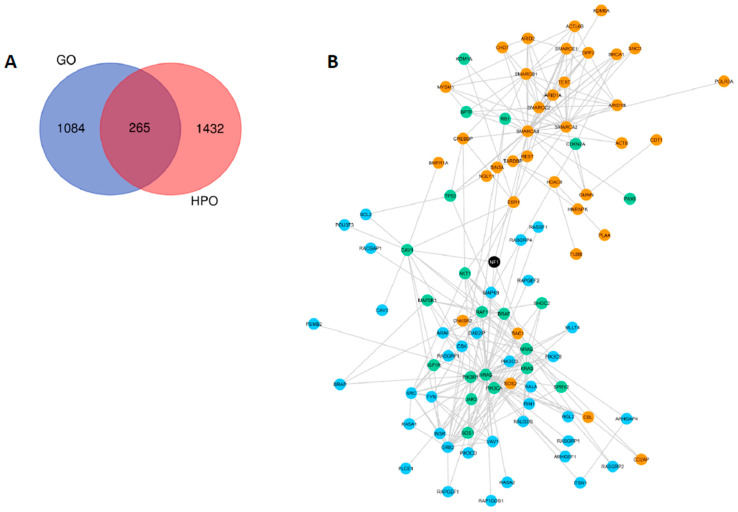
NF1 Gene Ontology (GO) and NF1 Human Phenotype Ontology (HPO) results. (**A**) Venn diagram showing in red genes related with neurofibromatosis type 1 exclusively found by the HPO project in the selected ontologies for NF1; genes exclusively found by the GO consortium using the ontologies selected for NF1 are shown in blue; and in purple genes shared by both GO and HPO analysis. (**B**) A combined network using the STRING tool using the 1697 genes selected exclusively in HPO (orange nodes); 1449 genes selected exclusively in GO (blue nodes); and 265 genes observed in both HPO and GO (green nodes). The network shows only direct protein–protein interactions with NF1 (first neighbors).

**Figure 3 cancers-12-02416-f003:**
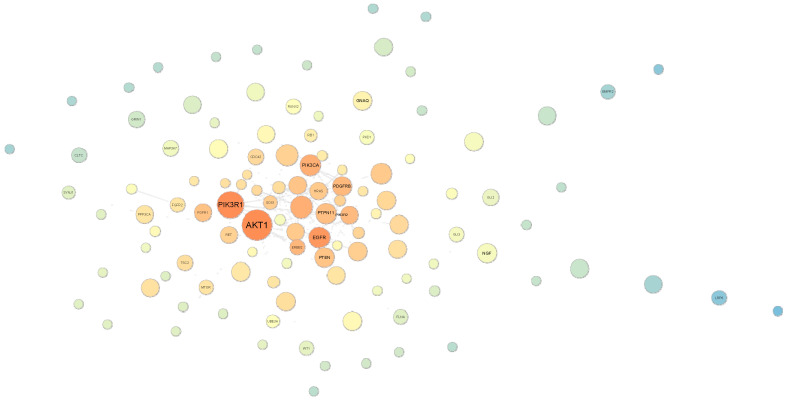
Betweenness centrality and closeness analysis of the STRING network previously generated in Figure 2b using GO and HPO. Large nodes have a more central role in communication among other nodes (hub/hub-like nodes), i.e., more connections. The darker orange the nodes are, the faster information flows towards the central node; i.e., they have the potential to impact the whole network even when having few connections (bottleneck nodes). Thus, nodes can be visualized in four categories: (1) large/dark-orange nodes = hub/bottleneck nodes; (2) large/blue nodes = hub/non-bottleneck nodes; (3) small/dark-orange nodes = non-hub/bottleneck nodes; and (4) small/blue nodes = non-hub/non-bottleneck hubs. NF1 is represented by the yellow node on the right side of AKT1.

**Figure 4 cancers-12-02416-f004:**
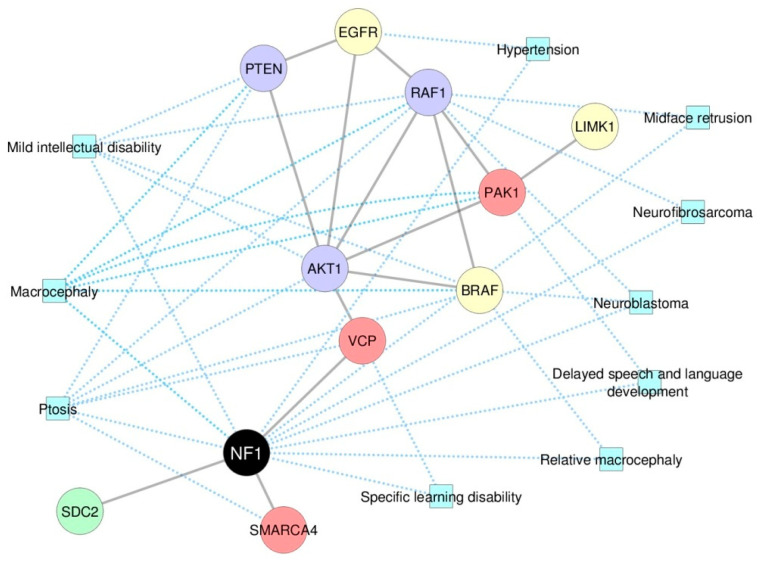
A complex network evaluation comprising the 10 candidate phenotype-modifier genes and the NF1 phenotypes they are related to. The ten candidate NF1 phenotype-modifier genes suggested by our analysis are represented with the NF1 phenotypes they are related. Yellow nodes: genes selected with the forward genetics strategy; red nodes: genes selected with the reverse genetics strategy; purple nodes: genes selected by forward and reverse genetics strategies; green node: gene selected by evaluating the differential gene expression in the complex network. Blue squares: phenotypes provided by the PhenomeScape tool, according to the associations presented in the HPO database.

**Figure 5 cancers-12-02416-f005:**
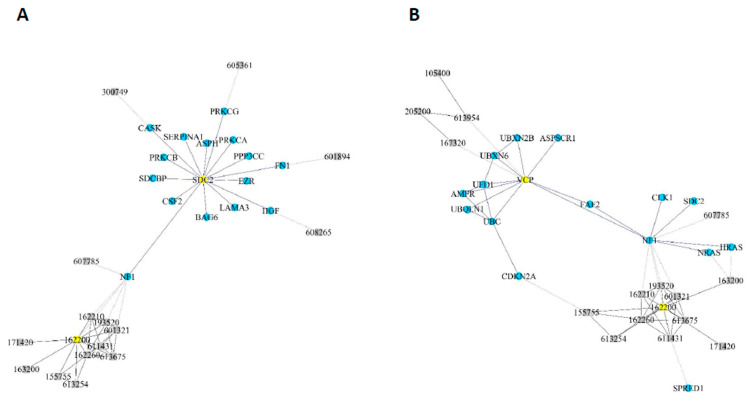
Random walk analysis. The minimum steps (interactions) between the selected genes (nodes) and the neurofibromatosis 1 phenotype were calculated. Two analyses are shown: (**A**) one for *SDC2*; and (**B**) one for *VCP*.

**Table 1 cancers-12-02416-t001:** Potential phenotype-modifier genes selected in this study. Characterization of the 10 potential phenotype-modifier genes and the approaches used for their selection.

Gene (OMIM)	Aliases	Cytogenetic Location	Summary	PINOT ^1^	STRING ^2^	BioGrid	Human Interactome	HPO	GO	Phenome Scape	Direct Strategy	Forward Strategy
*AKT1* (*164730)	*AKT*, *CWS6*, *PKB*, *PKB-ALPHA*, *PRKBA*, *RAC*, *RAC-ALPHA*	14q32.33	Serine/threonine kinase - development of the human nervous system; mediator of growth factor-induced neuronal survival; can inactivate components of apoptosis			X		X	X		X	X
*BRAF* (164757)	*NS7*; *B-raf*; *BRAF1*; *RAFB1*; *B-RAF1*	7q34	Serine/threonine kinase - role in regulating the MAP kinase/ERK signaling pathway						X		X	
*EGFR* (131550)	*ERBB*, *ERBB1*, *HER1*, *NISBD2*, *PIG61*, *mENA*	7p11.2	Cell surface protein - acts as a receptor for members of the epidermal growth factor family which induces cell proliferation	X		X			X		X	
*LIMK1* (601329)	*LIMK*; *LIMK-1*	7q11.23	Serine/threonine kinase - regulates actin polymerization; it is ubiquitously expressed during development; associated with cytoskeletal structure						X		X	
*PAK1* (602590)	*IDDMSSD*; *p65-PAK*; *PAKalpha*; *alpha-PAK*	11q13.5-q14.1	Serine/threonine kinase - cytoskeleton reorganization and nuclear signaling; regulates cell motility and morphology; essential for the RAS-induced malignant transformation					X				X
*PTEN* (601728)	*BZS*; *DEC*; *CWS1*; *GLM2*; *MHAM*; *TEP1*; *MMAC1*; *PTEN1*; *PTENbeta*	10q23.31	Phosphatidylinositol-3,4,5-trisphosphate 3-phosphatase that functions as a tumor suppressor	X		X		X	X		X	X
*RAF1* (164760)	*NS5*; *CRAF*; *Raf-1*; *c-Raf*; *CMD1NN*	3p25.2	MAP3 kinase - involved in the cell division cycle, apoptosis, cell differentiation and cell migration			X		X	X		X	X
*SDC2* (142460)	*HSPG*; *CD362*; *HSPG1*; *SYND2*	8q22.1	Syndecan proteoglycan protein - mediates cell binding, cell signaling, and cytoskeletal organization	X	X	X	X			X		
*SMARCA4* (603254)	*BRG1*; *CSS4*; *SNF2*; *SWI2*; *MRD16*; *RTPS2*; *BAF190*; *SNF2L4*; *SNF2LB*; *hSNF2b*; *BAF190A*; *SNF2-beta*	19p13.2	Part of the large ATP-dependent chromatin remodeling complex required for transcriptional activation of genes normally repressed by chromatin	X	X			X				X
*VCP* (601023)	*p97*; *TERA*; *CDC48*	9p13.3	Plays a role in protein degradation, intracellular membrane fusion, DNA repair and replication, regulation of the cell cycle, and activation of the NF-kappa B pathway	X	X	X	X	X				X

^1^ PINOT = Protein Interaction Network Online Tool; ^2^ STRING = Search Tool for Recurring Instances of Neighboring Genes.

**Table 2 cancers-12-02416-t002:** Variants reported in gnomAD for the 10 candidate genes. The total numbers and percentages of variants are presented according to their annotations.

GENE	All	Missense	Synonymous	Splice Site	Frameshift	Inframe Del/Ins	Intronic	Nonsense	Stop Lost	Start Lost	5′UTR	3′UTR
*AKT1*	932	166 (17.81%)	155 (16.63%)	80 (8.58%)	3 (0.32%)	4 (0.43%)	435 (46.67%)	1 (0.11%)	0	0	36 (3.86%)	52 (5.58%)
*BRAF*	1073	230 (21.44%)	170 (15.84%)	56 (5.22%)	1 (0.09%)	8 (0.75%)	561 (52.28%)	2 (0.19%)	1 (0.09%)	0	13 (1.21%)	31 (2.89%)
*EGFR*	2000	682 (34.10%)	387 (19.35%)	117 (5.85%)	15 (0.75%)	2 (0.10%)	867 (43.35%)	16 (0.80%)	1 (0.05%)	0	9 (0.45%)	104 (5.20%)
*LIMK1*	1133	322 (28.42%)	214 (18.89%)	58 (5.12%)	5 (0.44%)	1 (0.09%)	469 (41.39%)	3 (0.26%)	1 (0.09%)	0	44 (3.88%)	16 (1.41%)
*PAK1*	774	128 (16.54%)	121 (15.63%)	58 (7.49%)	4 (0.52%)	6 (0.78%)	401 (51.81%)	5 (0.65%)	1 (0.13%)	0	8 (1.03%)	42 (5.43%)
*PTEN*	456	83 (18.20%)	77 (16.89%)	17 (3.73%)	5 (1.10%)	0	223 (48.90%)	5 (1.10%)	0	0	28 (6.14%)	18 (3.95%)
*RAF1*	1005	264 (26.27%)	145 (14.43%)	65 (6.47%)	4 (0.40%)	2 (0.20%)	490 (48.76%)	7 (0.70%)	0	2 (0.20%)	8 (0.80%)	18 (1.79%)
*SDC2*	335	102 (30.45%)	52 (15.52%)	18 (5.37%)	5 (1.49%)	3 (0.90%)	114 (34.03%)	0	0	5 (1.49%)	22 (6.57%)	14 (4.18%)
*SMARCA4*	2575	473 (18.37%)	551 (21.40%)	164 (6.37%)	4 (0.16%)	22 (0.85%)	1318 (51.18%)	0	1 (0.04%)	0	9 (0.35%)	33 (1.28%)
*VCP*	928	135 (14.55%)	191 (20.58%)	72 (7.76%)	1 (0.11%)	1 (0.11%)	488 (52.59%)	0	0	0	20 (2.16%)	20 (2.16%)

**Table 3 cancers-12-02416-t003:** Germline variants submitted to ClinVar database for each candidate gene. The number of variants is presented according to its classification. Related syndromes and other relevant conditions for NF1 phenotype are also summarized.

GENE	Classification *					Related Syndromes ***	Other Relevant Reported Conditions
	All **	B/LB	P/LP	CI	VUS		
*AKT1*	182	94	8	4	76	Cowden, Proteus	E17K variant was associated with 22 conditions, including breast cancer
*BRAF*	334	125	80	12	117	Cardiofaciocutaneous, Dandy-Walker malformation, LEOPARD, PHACE, Noonan	Astrocytoma, glioma
*EGFR*	199	54	4	7	134	Cowden, not otherwise specified (NOS) hereditary cancer	Cerebral arteriovenous malformation, inflammatory skin and bowel disease
*LIMK1*	36	36	0	0	0	-	-
*PAK1*	6	2	3	-	1	-	intellectual developmental disorder with macrocephaly, seizures, and speech delay
*PTEN*	1567	367	510	22	668	Bannayan-Riley-Ruvalcaba, Cowden, Hereditary breast and ovarian cancer, NOS Hereditary cancer-predisposing, Proteus-like	Macrocephaly/autism, Phophatase and Tensin (PTEN) Homolog hamartoma tumor
*RAF1*	412	155	43	17	197	LEOPARD, Noonan	Chordoma, retinoblastoma, and
*SDC2*	-	-	-	-	-	-	leri pleonosteosis are reported in patients carrying the copy number gain of 8q22.1, which includes *SDC2*
*SMARCA4*	2310	980	95	81	1154	Coffin-Siris, NOS Hereditary cancer-predisposing, Rhabdoid tumor predisposition	Craniopharyngioma, intellectual deficiency, medulloblastoma, neurodevelopmental disorder, neuroblastoma
*VCP*	170	64	18	10	78	-	Amyotrophic lateral sclerosis, paget disease, Charcot-Marie-Thoth disease

* LB = likely benign; B = benign; LP = likely pathogenic; P = pathogenic; CI = conflicting interpretation; VUS = variant of uncertain significance. ** Unprovided interpretations and drug response variants were not considered. *** Syndromes/conditions reported for benign/likely benign variants, drug response variants, and not-interpreted variants were not considered.

**Table 4 cancers-12-02416-t004:** List of genes and proteins previously described as NF1 phenotype modifiers in the literature.

Genes/Proteins	Consequence	Methodology Aspects	Reference
*ADCY8*	Genetic polymorphisms in *ADCY8* are correlated with glioma risk in NF1 in a sex-specific manner, elevating risk in females while reducing risk in males	- Genotyping of NF1 patients using Affymetrix whole-genome human SNP array - Primary astrocyte cultures from NF1-CKO mice and treatment with dideoxyadenosine to induce ADCY inhibition - cAMP regulator expression with qPCR and ELISA	Warrington et al. 2015 [33]
*ANRIL* allele T of SNP rs2151280	Higher number of plexiform neurofibromas; rs2151280 reduced *ANRIL* transcript levels	- High-resolution array comparative genomic hybridization (aCGH) of PNFs from NF1 patients	Pasmant et al. 2011 [22]
*ATP6V0B* SNP rs7161 *DPH2* SNP rs4660761 *MSH6* SNP rs1800934	*ATP6V0B* is associated with melanosome biology rs7161 and rs4660761 associated with café-au-lait macule (CALM) count; rs1800934 associated with development an NF1-like phenotype	- Lymphoblastoid cell lines with NF1-associated phenotypes - Gene expression (microarray and qPCR) - Sequencing of genes with incresased expression in patients with high count CALM - Meta-analysis	Pemov et al. 2014 [19]
*CRLF3, ADAP2, RNF135, UTP6, SUZ12, OMG, LRRC37B, EVI2A, EVI2B, RAB11FIP4, RAB11FIP3, TEFM, ATAD5, CORPS, NF1* large 17q11 deletions encompassing the entire NF1 locus and neighboring genes	Dysmorphic features, learning disabilities, cardiovascular malformations, childhood overgrowth, a higher tumor burden and earlier onset of benign neurofibromas, and probably, a higher incidence of malignant peripheral nerve sheath tumors (MPSTs)	-MLPA, breakpoint-spanning PCR and FISH in NF1 deletion patients	Mautner et al. 2010 [13]
CXCR4 and its ligand, CXCL12	Highly expressed in mouse models of NF1-deficient MPNSTs, but not in normal precursor cells; Suppression of CXCR4 activity decreases MPNST cell growth in culture and inhibits tumorigenesis in allografts and in spontaneous genetic mouse models of MPNST; Demonstrated conservation of these activated molecular pathways in human MPNSTs	- NF1-deficient skin-derived precursor models (SKPs) and gene expressuion microarray (normal SKPs; pretumorigenic SKPs with either Nf1 deletion or Nf1 and p53 deletion) - qPCR, westerblot and IHC of CXCR4 and CXCL12 - CXCR4 shRNA for knockdown in SKP MPNST cells - Tissue microarray from plexiform neurofibromas in NF1 patients harboring MPNSTs and MPNSTs samples from NF1 and sporadic patients - CXCR4 cDNAs sequencing	Mo et al. 2013 [34]
*GDNF* R93W germiline variant and maternally inherited *NF1* mutation	Congenital megacolon development	- Investigation of a family carrying variants in *GDNF* and *NRTN* genes with cutaneous manifestations of NF1 and megacolon - Haploinsufficient animal models for *Nf1* and *Trp53* that developed MPNSTs	Bahuau et al. 2001 [35]
miR-34a	Down-regulation of miR-34a founded in most MPNSTs compared to neurofibromas; The p53 inactivation and subsequent loss of expression of miR-34a may contribute to MPNST development	- Microarray of MPNSTs, neurofibromas, Schwannonas, and synovial sarcomas - MPNST cell lines to check for miR-34a and other p53-dependent miRNAs by qRT-PCR after overexpressing wild-type p53	Subramanian et al. 2010 [36]
miR-21	Important in MPNST tumorigenesis and progression through its target, PDCD4	- Global miRNA expression profiling of MPNSTs and neurofibromas - qPCR of differentially expressed miRNAs in MPNSTs, 11 NFs, and 5 normal nerves and MPNST cell lines - Knockdown of miR-21 in MPNST cells	Itani et al. 2012 [37]
miR-204	Down-regulation of miR-204 contributes to development and tumor progression of MPNSTs	- Global miRNA expression profiling of MPNSTs and benign NF1 neurofibroma tissues - qPCR of differentially expressed in tumor tissues and MPNST cell lines - Lentiviral system for miR-204 trasnfection in NF1 and non-NF1 MPNST cell lines - Non-NF1 MPNST cells Xenograft	Gong et al. 2012 [38]
*MSH2, MSH6, MSH3, MLH1, PMS2*	Phenotype overlapping between NF1 and Constitutional mismatch repair deficiency (CMMRD) Association with rare childhood malignancies	- Literature review about co-occurrence of symptoms and variants in genes associated with CMMRD and NF1	Wimmer, Rosenbaum and Messiaen 2017 [39]
*SDHB*	Cause gastrointestinal stromal tumor (GISTs)	- SDHB expression by immunohistochemically in NF1-associated GISTs	Wang, Lasota and Miettinen 2011 [40]
Serotonin receptor *5′UTR* 5-HT_6_ - HTR6 protein	Disrupting HTR6-neurofibromin interaction prevents agonist-independent HTR6-operated cAMP signaling in the prefrontal cortex, an effect that might underlie neuronal abnormalities in NF1 patients; 5-HT_6_ receptor may be considered as a potentially therapeutic target to correct some NF1-related cognitive deficits	- Nf1+/− heterozygote mice - HEK-293T and NG108-15 cell lines - Immunoprecipitation followed by Western blot analysis	Deraredj Nadim et al. 2016 [41]
*SPRED1* nonsense, frameshift and missense mutations	Complete *SPRED1* inactivation is needed to generate CALMs	- GWAS in unaffected and affected individuals. - *SPRED1* cDNA sequencing - Melanocyte cell culture from normal skin and CALM of NF1 patient -Mouse embryonic fibroblasts	Brems et al. 2007 [42]
*TERT* mRNA and telomerase activity	Telomere dysfunction may play a role in driving genomic instability and clonal progression in NF1-associated MPNST	- High-resolution Single Telomere Length Analysis (STELA) of cutaneous and diffused plexiforme neurofibromas, and MPNSTs	Jones et al. 2017 [43]

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
