# Peer review of "Systems Biology Approaches Reveal Potential Phenotype-Modifier Genes in Neurofibromatosis Type 1"

_cancers, 2020, doi:10.3390/cancers12092416_

Round 1
Reviewer 1 Report
This original article (manuscript cancers-867098) by Woycinck Kowalski et al. describes an in silico assessment of potential modifier genes of the neurofibromatosis type 1 (NF1)-associated phenotype.
The authors propose an original approach to suggest NF1 modifier genes. However, this analysis is based on the selection of terms and keywords, which is arbitrary and could bias the analysis.
The authors also review previous studies dealing with genotype-phenotype correlation and modifier genes in NF1. It would be interesting to describe in more detail the previous works that led to suggest that there are modifier genes of the NF1 phenotype.
Major remarks:
- The choice of the GO and HPO terms (in section 2.1) is critical as it determines the analysis; however, there is little justification for the selection of individual terms. The arguments that led to the selection of these items should be added because it determines the results.
- GO terms: why chose “Regulation Of Cell-Matrix Adhesion”? Other relevant terms would have been selected; e.g. «GTPase activator activity», «neural tube development», «wound healing» (possible role of inflammation induced during the healing process in the formation of neurofibromas), «negative regulation of osteoclast differentiation», «negative regulation of fibroblast proliferation», «negative regulation of astrocyte differentiation», «negative regulation of oligodendrocyte differentiation» et «forebrain morphogenesis»
- HPO terms: why chose: «ptosis», «Sparse and thin eyebrow», «Midface retrusion», «Hypertrophic cardiomyopathy»; these terms mostly relate to the Noonan/Noonan-like phenotype. In this case, why didn't you also selected additional terms related to the Noonan phenotype like “Hypertelorism“, «Abnormality of the cardiovascular system» and « Pulmonic stenosis». The following terms might also have been relevant: «Neuroblastoma», «Pectus excavatum of inferior sternum», «Superior pectus carinatum», «Abnormality of the thorax», «Inguinal freckling», «Lisch nodules», «Cafe-au-lait spot» and «Short stature».
-The introduction could be expanded with a deeper description of previous genotype-phenotype correlations in NF1. Indeed, there are certain particular mutations associated with specific phenotypes; it would be interesting to present them in the introduction. Subsequently, the authors consider that there is evidence for modifier genes in NF1; they could describe them more and include the supplemental table S4 in the manuscript.
Additional remarks:
In Section 2.2, it says: “NF1 being a highly connected protein in the network evaluated, it presented low levels of betweenness and closeness centrality, as can be observed by the node size (small) and color (light yellow, compared to the dark orange elements).“ However, NF1 is not shown in figure 3 which could be more informative and could show all the genes, like in figures S1 and S2. The NF1 gene should be highlighted in the figure.
In Section 2.4., the strategy of analyzing NF1-mutated versus wild-type tissues for the identification of putative modifier genes should be more explained in the discussion, as it could show different bias. Especially since the results are not the same using the GEO and TCGA approaches.
In Section 2.4., it says: “The logFC and adjusted P-value, after the FDR correction, for the seven candidate genes are available in Table S3“, There are 10 modifier genes and not 7.
In Section 2.4., Figures S3 to S7. Authors should explain explain why NF1 is sometimes written in red (down-regale), sometimes in green (over-expressed), and sometimes absent from the network.
Author Response
Comments and Suggestions for Authors
This original article (manuscript cancers-867098) by Woycinck Kowalski et al. describes an in silico assessment of potential modifier genes of the neurofibromatosis type 1 (NF1)-associated phenotype.
-The authors propose an original approach to suggest NF1 modifier genes. However, this analysis is based on the selection of terms and keywords, which is arbitrary and could bias the analysis.
Answer: We thank the reviewer for this important observation. Our strategy to avoid bias based on the selection of terms and keywords was better explained in the Methods section 4.1, lines 471-488, in the revised manuscript version.
-The authors also review previous studies dealing with genotype-phenotype correlation and modifier genes in NF1. It would be interesting to describe in more detail the previous works that led to suggest that there are modifier genes of the NF1 phenotype.
Answer: We included more details about the previous works that suggested NF1 phenotype modifiers in Table 4 (Discussion section, line 260). In the previous version of the manuscript, this table was named Table S4. In the revised manuscript version, the table was incorporated in the main text.
Major remarks:
- The choice of the GO and HPO terms (in section 2.1) is critical as it determines the analysis; however, there is little justification for the selection of individual terms. The arguments that led to the selection of these items should be added because it determines the results.
GO terms: why chose “Regulation Of Cell-Matrix Adhesion”? Other relevant terms would have been selected; e.g. «GTPase activator activity», «neural tube development», «wound healing» (possible role of inflammation induced during the healing process in the formation of neurofibromas), «negative regulation of osteoclast differentiation», «negative regulation of fibroblast proliferation», «negative regulation of astrocyte differentiation», «negative regulation of oligodendrocyte differentiation» et «forebrain morphogenesis»
HPO terms: why chose: «ptosis», «Sparse and thin eyebrow», «Midface retrusion», «Hypertrophic cardiomyopathy»; these terms mostly relate to the Noonan/Noonan-like phenotype. In this case, why didn't you also selected additional terms related to the Noonan phenotype like “Hypertelorism“, «Abnormality of the cardiovascular system» and « Pulmonic stenosis». The following terms might also have been relevant: «Neuroblastoma», «Pectus excavatum of inferior sternum», «Superior pectus carinatum», «Abnormality of the thorax», «Inguinal freckling», «Lisch nodules», «Cafe-au-lait spot» and «Short stature».
Answer: We thank the reviewer appointment and agree that GO and HPO terms are critical to the analysis. We provided a more complete explanation and arguments that lead to the choice of the GO and HPO terms in the Methods section 4.1, lines 471-488 and in the Discussion section, lines 294-308 of the revised manuscript. Briefly, gene ontology (GO) analysis is usually applied to identify enriched genes in a differential expression set. However, our aim was to use this strategy for a correct selection of potential phenotype modifier genes. This was an important step in our analysis, especially because NF1 is important in several molecular mechanisms. If not filtered, our analysis could be later compromised by evidencing genes that are not so deeply associated to neurofibromatosis type 1, but which are more frequently studied in cancer (i.e. TP53 and development genes). This strategy has also been applied in studies that do not identify (or do not have access to) differentially expressed genes [PMID 29328373 and 32038721]. For Human Phenotype Ontology (HPO), we are also aware that many ontologies were not selected, but our workflow was based on a deep phenotyping strategy (computational analysis of detailed, individual clinical abnormalities) [PMID 30476213], according to the heterogeneity of neurofibromatosis type 1 visualized in our patients. HPO has been widely allied to next-generation sequencing (NGS) in the diagnosis of conditions with significant phenotypic heterogeneity [PMID 30651242], such as neurofibromatosis type 1. We also aimed to avoid ontologies related to congenital anomalies outside the NF1 spectrum of phenotypes, especially the ones that could have led to embryo lethality or severe impairments (major malformations) that would have been diagnosed before NF1. Embryo development is a critically, stepwise controlled process, that can be disrupted by genetic or environmental factors, such as maternal infections or exposures [PMID 30095839]. Since the data on NF1 expression in the embryonic period is scarce, and we do not have information about the maternal genome or environment, we focused on the functional anomalies (more related to the fetal period) that are better characterized in neurofibromatosis type 1. It is important to highlight that we followed a guide for a correct selection of ontologies [PMID 26867217]. We also considered the hierarchical relationships of Gene Ontologies [PMID 31127124]; hence, some ontologies were not selected because there was an upper term in the hierarchy that encompassed these ontologies. Finally, Noonan/Noonan-like features are variably present in NF1 patients, with a low frequency; for example, a well described genotype-phenotype correlation in NF1 is the presence of a missense variant p.(p.Arg1809Cys), associated with developmental delay and/or learning disabilities, pulmonic stenosis, and Noonan-like features, and for this reason we selected ontologies related to all Noonan-like features (the therms that do not appear in the selection were part of a upper term in the hierarchy).
-The introduction could be expanded with a deeper description of previous genotype-phenotype correlations in NF1. Indeed, there are certain particular mutations associated with specific phenotypes; it would be interesting to present them in the introduction. Subsequently, the authors consider that there is evidence for modifier genes in NF1; they could describe them more and include the supplemental table S4 in the manuscript.
Answer: We thank the reviewer suggestions. The previously described genotype-phenotype correlations in NF1 and the characterization of modifier genes were included in the Introduction section, lines 50-57 and lines 60-64 in the revised manuscript version, respectively. Novel references were included and are highlighted in red. The supplementary table S4 was revised, more details were included about the studies that described NF1 phenotype modifiers and it was incorporated in the manuscript as Table 4 (Discussion section, line 260).
Additional remarks:
-In Section 2.2, it says: “NF1 being a highly connected protein in the network evaluated, it presented low levels of betweenness and closeness centrality, as can be observed by the node size (small) and color (light yellow, compared to the dark orange elements).“ However, NF1 is not shown in figure 3 which could be more informative and could show all the genes, like in figures S1 and S2. The NF1 gene should be highlighted in the figure.
Answer: We thank to the reviewer for the suggestion. In Figure 3, NF1 is quite small but it is located besides AKT1. For a better comprehension, we put red arrows pointing to NF1 in Figures 3, S1 and S2. In regards to the suggestion of showing all the genes in Figure 3, we believe it is important, but the network consists of 1,561 nodes, making it very difficult to evaluate its interactions. The complete figure is below, for your appreciation (please see the attachment with the figure). Nevertheless, we added the information about the number of nodes in the text (Results section 2.2, lines 124-126).
- In Section 2.4., the strategy of analyzing NF1-mutated versus wild-type tissues for the identification of putative modifier genes should be more explained in the discussion, as it could show different bias. Especially since the results are not the same using the GEO and TCGA approaches.
Answer: We appreciate this suggestion and included a sentence pointing to this difference in the Discussion section, lines 270-272 of the revised manuscript. We also included in lines 272-274 a possible explanation for this bias.
-In Section 2.4., it says: “The logFC and adjusted P-value, after the FDR correction, for the seven candidate genes are available in Table S3“, There are 10 modifier genes and not 7.
Answer: We thank the reviewer for the careful examination. The information was corrected in the Results section 2.4, lines 168-169. There are ten modifier genes and seven tumors. We are sorry for the typing mistake.
-In Section 2.4., Figures S3 to S7. Authors should explain why NF1 is sometimes written in red (down-regale), sometimes in green (over-expressed), and sometimes absent from the network.
Answer: We appreciate the reviewer suggestion. As expected from other literature reports, NF1 is downregulated (low expression) in knockdown and knockout studies, and upregulated (overexpressed) in malignant tumors when compared to benign neurofibromas. NF1 is absent from the network when its expression is not significantly altered in the analysis performed. We specified this information in the revised version of the manuscript (Results section 2.4, lines 179-183) and in the S4 and S7 figure captions.

Reviewer 2 Report
Very interesting & novel work. In line 286-289, I would not consider breast cancer to be a rare manifestation of NF1 patients & would prefer that line to be altered.
Author Response
Comments and Suggestions for Authors
Very interesting & novel work. In line 286-289, I would not consider breast cancer to be a rare manifestation of NF1 patients & would prefer that line to be altered.
Answer: We thank to the reviewer observation and changed the sentence to “The link between breast cancer susceptibility and NF1 alterations was already established [18]”, in the Discussion section, line 360 of the revised manuscript.
Reviewer 3 Report
The authors present a new approach to the study of the phenotypic variability of Neurofibromatosis I by combining network analysis of genes and phenotypic features through the use of existing public databases.
THe work is interesting and clearly described, the authors try to combine several analyses to give reliability to the results, but a merely computational approach seems too dependent from the input variables selected by the authors
Moreover, the authors should point out that some mechanisms of variability are already known (i.e. the second somatic hit in the NF1 gene in many tissues) and are not dependent (or they are marginally dependent) from supposed modifiers
In conclusion, the article could be read like a proof-of-concept application of system biology to neurofibromatosis type I, but the results must be interpreted very cautiously and the authors should really stress this concept throughout the paper
Author Response
Reviewer 3
Comments and Suggestions for Authors
The authors present a new approach to the study of the phenotypic variability of Neurofibromatosis I by combining network analysis of genes and phenotypic features through the use of existing public databases.
-The work is interesting and clearly described, the authors try to combine several analyses to give reliability to the results, but a merely computational approach seems too dependent from the input variables selected by the authors
Answer: We thank to the reviewer for the comments. Gene ontology (GO) analysis was an important step in our study, especially because NF1 is important in several molecular mechanisms. If not filtered, our analysis could be later compromised by evidencing genes that are not so deeply associated to neurofibromatosis type 1, but which are more frequently studied in cancer (i.e. TP53 and development genes). For Human Phenotype Ontology (HPO), our workflow was based on a deep phenotyping strategy (computational analysis of detailed, individual clinical abnormalities) [PMID 30476213], according to the heterogeneity of neurofibromatosis type 1 visualized in our patients. HPO has been widely allied to next-generation sequencing (NGS) in the diagnosis of conditions with significant phenotypic heterogeneity [PMID 30651242], such as neurofibromatosis type 1. It is important to highlight we followed a guide for a correct selection of ontologies [PMID 26867217]. For a better comprehension of the strategies used, we added additional information and the references here mentioned in the Introduction section, lines 74-75 and 78-82; in the Discussion section, lines 294-308; and in the Materials and Methods section, lines 471-488 of the revised version of the manuscript.
- Moreover, the authors should point out that some mechanisms of variability are already known (i.e. the second somatic hit in the NF1 gene in many tissues) and are not dependent (or they are marginally dependent) from supposed modifiers
Answer: We agree with the reviewer comment and briefly discussed about NF1 second somatic hits in the Discussion section, lines 249-254.
- In conclusion, the article could be read like a proof-of-concept application of system biology to neurofibromatosis type I, but the results must be interpreted very cautiously and the authors should really stress this concept throughout the paper
Answer: We thank to the reviewer for the very important observation. This systems biology analysis is an exploratory evaluation for the suggestion of new candidate phenotype modifiers in neurofibromatosis type 1, and these results must be further evaluated and experimentally validated before a clinical application. To better explain this concern, we added a sentence in the manuscript Discussion section, lines 460-461 and in the Conclusion section, lines 568-571 of the revised manuscript.